# Proteomic Analysis of Mouse Brain Subjected to Spaceflight

**DOI:** 10.3390/ijms20010007

**Published:** 2018-12-20

**Authors:** Xiao Wen Mao, Lawrence B. Sandberg, Daila S. Gridley, E. Clifford Herrmann, Guangyu Zhang, Ravi Raghavan, Roman A. Zubarev, Bo Zhang, Louis S. Stodieck, Virginia L. Ferguson, Ted A. Bateman, Michael J. Pecaut

**Affiliations:** 1Department of Basic Sciences, Division of Biomedical Engineering Sciences, Loma Linda University School of Medicine, Loma Linda, CA 92354, USA; xmao@llu.edu (X.W.M.); dgridley@llu.edu (D.S.G.); 2Department of Biochemistry, Loma Linda University School of Medicine, Loma Linda, CA 92350, USA; lsandberg@llu.edu (L.B.S.); cherrmann@llu.edu (E.C.H.); guangyuzhang@llu.edu (G.Z.); 3Department of Pathology and Human Anatomy, Loma Linda University School of Medicine, Loma Linda, CA 92350, USA; rraghavan@llu.edu; 4Department of Medical Biochemistry and Biophysics, Biomedicum, Karolinska Institutet, SE 17177 Stockholm, Sweden; Roman.Zubarev@ki.se (R.A.Z.); Bo.Zhang@ki.se (B.Z.); 5Department of Pharmacological and Technological Chemistry, I.M. Sechenov First Moscow State Medical University, Moscow 119991, Russia; 6BioServe Space Technologies, University of Colorado at Boulder, Boulder, CO 80303, USA; stodieck@colorado.edu (L.S.S.); Virginia.Ferguson@colorado.edu (V.L.F.); 7Department of Bioengineering, University of North Carolina at Chapel Hill, Chapel Hill, NC 27599, USA; tbateman@email.unc.edu

**Keywords:** brain, spaceflight, microgravity, proteomics

## Abstract

There is evidence that spaceflight poses acute and late risks to the central nervous system. To explore possible mechanisms, the proteomic changes following spaceflight in mouse brain were characterized. Space Shuttle Atlantis (STS-135) was launched from the Kennedy Space Center (KSC) on a 13-day mission. Within 3–5 h after landing, brain tissue was collected to evaluate protein expression profiles using quantitative proteomic analysis. Our results showed that there were 26 proteins that were significantly altered after spaceflight in the gray and/or white matter. While there was no overlap between the white and gray matter in terms of individual proteins, there was overlap in terms of function, synaptic plasticity, vesical activity, protein/organelle transport, and metabolism. Our data demonstrate that exposure to the spaceflight environment induces significant changes in protein expression related to neuronal structure and metabolic function. This might lead to a significant impact on brain structural and functional integrity that could affect the outcome of space missions.

## 1. Introduction

Long-term deep space missions expose astronauts to an environment that is characterized mainly by ultraviolet and ionizing radiation, microgravity, and physiological/psychological stressors. These conditions present a significant hazard to spaceflight crews during and after the course of mission activities. The hazards posed to normal tissues, such as the central nervous system (CNS), are not fully understood. The health risks of spaceflight-induced neuronal damage and potential neurodegenerative effects have long been a concern. Brain damage and degeneration can be promoted by many factors including aging, ischemia, fluctuation in oxygen tension, oxidative stress, and increased intraocular pressure.

There is some evidence that low-dose, space-relevant radiation induces changes in neuronal functions [1]. Microgravity induces intraocular pressure and vascular changes [2,3] and promotes apoptosis of astrocytes [4]. Spaceflight also induces cognitive and perceptual motor performance deterioration under stress [5]. Studies have shown that exposure to spaceflights has a strong impact on metabolic and stress response [6]. Collective evidence indicates that exposure to stressful spaceflight environments might induce changes in brain neuronal structure and function. However, the pathophysiological consequences and role of cellular mechanisms of stress stimuli, especially those stemming from the spaceflight environment, in facilitating brain damage and neurodegeneration have been studied less and remain unclear.

Gray matter consists of neurons (i.e., it contains the cell bodies, dendrites and axon terminals of neurons), nerve fibers, astrocytes, microglia, and capillaries. Gray matter is closely associated with the functional domains of performance, locomotion, learning, memory and coordination. On the other hand, white matter consists mostly of oligodendroglial cells, myelinated axons and capillaries. White matter allows communication to and from gray matter areas, and between gray matter and the other parts of the body. It functions by transmitting the information from the different parts of the body towards the cerebral cortex. It modulates the distribution of action potentials, acting as a relay and coordinating communication between different brain regions [7]. Changes in gray matter are known to be primarily associated with Alzheimer’s disease and other neurodegenerative diseases, with secondary effects on the white matter [8]. The deficits range from language ability to delayed memory and visuospatial construction. Disrupted white matter organization has been linked to poorer motor performance [9]. Studies have shown altered expressions of a number of genes and proteins involving a wide spectrum of biological functions following exposure to space environments. These alterations induced distinct changes specific to the regions of the brain [10]. Regional difference in stress response was also documented following simulated microgravity in human brain gray matter and white matter [11].

The purpose of the present investigation was to study spaceflight condition-induced changes in protein expression profiles in mouse gray matter and to compare these changes to those in white matter regions. Our unique data might provide new insights and improve risk assessment for future long-term space travel.

## 2. Results

There were nine and 17 proteins that were significantly altered after spaceflight in the white (Table 1) and gray (Table 2) matter, respectively (*p* < 0.05, log fold change >1.0 or <−1.0). In general, proteins that were significantly altered were upregulated in both areas of the brain. However, there was no overlap between the brain area data sets. If log fold change constraints were reduced to >0.5, the number of proteins increased to 16 and 25 for white and gray matter, respectively.

There were no significant changes in canonical pathways or upstream regulators in the pathway analysis for either the white or gray matter proteins. However, there were strong trends for changes in functionally related proteins in both brain regions (Table 3). In the white matter, there was a strong trend for a downregulation of functions related to the formation of cellular protrusions (*Z* = −1.98). In the gray matter, there was a significant downregulation of functions related to overall organismal death and organ degeneration (*Z* < −2.0). There were also strong trends for downregulation in cellular and neural degenerative functions (*Z* = −1.98). Interestingly, there was a significant upregulation in functions related to viral infection (*Z* > 2.0).

## 3. Discussion

While there do not appear to be enough significantly different proteins in either white or gray matter in this analysis to appear as a significantly activated canonical pathway, there still appear to be common functional themes: (1) synaptic function, (2) intracellular communication, (3) metabolism, (4) oxidative stress and tissue damage responses, and (5) activation of catecholamines.

### 3.1. Synaptic Function: Plasticity, Vesicles and Dendritic Spines

In the white matter there were three proteins related to synaptic plasticity that were upregulated. Calcium voltage-gated channel auxiliary subunit α2δ1 (CACNA2D1) is intimately involved in calcium channel trafficking [12] and regulates excitatory synapse formation during development or after injury [13]. PTPRF interacting protein α 3 (PPFIA3, also known as Liprin-α-3) is typically found to be co-expressed with a variety of pre-synaptic proteins in neurons but has also been found in astrocytes [14,15]. It is thought to be involved in presynaptic plasticity and synaptic vesicle release, particularly in excitatory synapses [15,16]. Myosin VA (MYO5A) is an F-actin-based motor protein that is also important in the generation and movement of synaptic vesicles. It has been found in dendritic spines and synaptic vesicles and appears to be critical for synaptic plasticity and organelle transport (reviewed in Reference [17]).

Several proteins associated with synaptic function that are upregulated in the gray matter after flight involve vesicle formation, exocytosis and endocytosis. Syntaxin 1A (STX1A) is a soluble *N*-ethylmaleimide-sensitive fusion attachment protein receptor (SNARE) protein that is expressed in most neurons [18] and is a critical component of synaptic vesicle formation and exocytosis [19,20]. DNAJ heat shock protein family (Hsp40) member C5 (DNAJC5) is a pre-synaptic DNAJ C-class Hsp40 co-chaperone that is primarily expressed in the brain and retina (reviewed in [21]). It is part of a complex of proteins that resides on synaptic vesicles and chaperones pre-synaptic SNARE proteins, making it critical during repeated synaptic vesicle cycles [22]. Indeed, DNAJC5 knockout mice have progressive, age-dependent sensorimotor deficits and the protein appears to be critical to preventing pre-synaptic degeneration via deficits in endocytosis [21,23].

Upregulated within gray matter, dynamin 3 (DNM3) is expressed in the dendritic spines and is primarily associated with regulating synaptic vesicle endocytosis and recycling [24,25,26]. There are three isoforms of dynamin that share about 80% overall homology and have mostly redundant roles in clathrin-mediated endocytosis and membrane fission [24,27]. The SH3 domain containing GRB2-like 2, endophilin A1 (SH3GL2, also known as endophilin-1) is a potential tumor suppressor gene [28] that is highly expressed in the brain, particularly in presynaptic ganglion [29]. However, the primary function of this protein is to regulate clathrin-mediated endocytosis (reviewed in [30]). Finally, while not directly related to neuronal communication, ATPase H+ transporting V0 subunit A1 (ATP6V0A1) involves a form of endocytosis in microglia. This protein that was upregulated in gray matter, is involved with the merging of lysosomes and phagosomes during phagocytosis in the brain [31]. Interestingly, clathrin-mediated endocytosis was significantly and highly upregulated in the liver of these mice as well [32], suggesting a systemic response.

In addition to vesicle formation, several of the proteins upregulated in gray matter after spaceflight have been implicated in neurite and dendritic spine formation. Tyrosine 3-monooxygenase/tryptophan 5-monooxygenase activation protein epsilon (YWHAE) is believed to play a critical role during neuronal development and migration [33] and neurite formation during cortical development [34]. Over expression of this gene disrupts neurite formation through the microtubule binding protein, doublecortin [34]. Enolase 2 (ENO2) has also been shown to have neurotrophic activity [35] and is involved in cytoskeletal remodeling and neurite regeneration [36].

Similarly, both dynamin 3 (DNM3) [37] and SH3 domain containing GRB2-like 2, endophilin A1 (SH3GL2) [38] that were upregulated in gray matter are involved with dendritic spine morphogenesis and stability. Another upregulated protein in the gray matter is actinin alpha 1 (ACTN1). Expressed in the dendritic spines of the post-synaptic density (PSD) [39,40], this protein is an actin-crosslinking protein [41] that is involved with synaptic plasticity [39]. This is interesting because changes in dendrite activity might be related to synaptic plasticity [42,43].

Consistent with the upregulation of proteins related to neurite and dendrite growth is the downregulation of SEC22B (SEC22 homolog B, a vesicle trafficking protein) in the gray matter. Knocking down this protein using siRNA reduced neurite length had no impact on neuronal migration [44].

### 3.2. Intracellular Communication: Myelination and Protein/Organelle Transport

Another broad category impacted by spaceflight is involved in intracellular communication. Specifically, (1) axonal signaling that is “insulated” via myelin and (2) protein and organelle transport. Downregulated in the murine white matter, myelin basic protein (MBP) has an important role in the process of myelination of axons, particularly in the adhesion of myelin layers between cytosolic surfaces [45,46]. MBP is implicated in auto-immune responses within the human CNS, and is thought to be a target for T cell activity in multiple sclerosis and other demyelinating or degenerative disorders. Its reduction over an extended period is usually associated with glial inflammation activation and proliferation, leading to reactive astrocytosis [47,48].

Interestingly, three important factors found in oligodendrocytes and involved in myelin formation were upregulated in the gray matter: acyl-coenzyme A, a cholesterol acyltransferase 1 (ACAT1) [49]; 2′,3′-cyclic nucleotide 3′-phosphodiesterase (CNP) [50,51]; and neurofilament light (NEFL) [52,53].

Two components of protein/organelle transport were upregulated in the white matter. As stated previously, myosin VA (MYO5A) is important for organelle transport (reviewed in Reference [17]). Similarly, dynein light chain LC8-type 2 (DYNLL2, also known as DLC2), originally identified as part of the microtubule-based motor protein dynein [54], is involved with transporting mitochondria along the axons of neurons in response to local energy and metabolic requirements [55]. However, DYNLL2 has also been shown to have a variety of other targets including nNOS, post synaptic scaffolding proteins, and pro-apoptotic proteins [54].

However, there were also two factors involved in protein transport that were downregulated in white matter. Vacuolar protein sorting 35 ortholog (VPS35) is a component of the “cargo recognition complex” of the retromer complex responsible for the retrograde transport of proteins from endosomes to the trans-Golgi network or the plasma membrane [56,57]. Already mentioned above as an important component of myelin, MBP also interacts with the cytoskeleton and/or tight junctions, making it critical for communicating extracellular signaling to the inside of the cell [46]. Decreases in MBP have been associated with glial activation [58].

### 3.3. Metabolism: Glycolysis and Mitochondrial Function

Consistent with our previous results, in the liver and skin of the same mice in which we found that spaceflight had a major impact on metabolism [32,59,60], the next broad category involved the impact of spaceflight on the brain, including glycolysis and metabolism.

Two proteins involved in glycolysis or metabolism were altered by spaceflight in the white matter. This first one was upregulated. Phosphatidylinositol transfer protein α (PITPNA) is involved with coordinating lipid metabolism and signaling [61], transferring phospholipids out of the endoplasmic reticulum and into other membranes [62]. Interestingly, an increase in oxidative stress has been shown to cause a decrease in this protein, particularly in the brains of aged or Parkinson’s disease models [63]. This lack has been associated with neurodegenerative disease, which has been linked to changes in glucose homeostasis [64]. Glyceraldehyde-3-phosphate dehydrogenase (GAPDH) is an abundant enzyme in brain tissue that is critical to energy metabolism and glycolysis [65]. In conditions of oxidative stress, GAPDH activity is impaired, leading to cellular aging and apoptosis [65]. This enzyme can undergo sulfhydration, S-nitrosylation and oxidation, which, in turn, can lead to memory loss, apoptotic cell death and neurodegeneration, noted in ischemia and Alzheimer’s disease models [66,67,68].

In the gray matter, enolase 2 (ENO2) was upregulated after flight, also suggesting glycolysis may be impacted. ENO2 is a glycolytic enzyme [69] found in neurons, neuroendocrine cells and microglia [69,70]. However, one of the few proteins that were downregulated in the gray matter, SEC22B (SEC22 homolog B, vesicle trafficking protein), also complexes with SNARE proteins in the endoplasmic reticulum (ER) [44,71]. Knocking down this protein using siRNA had no impact on exocytosis [44]. Instead, SEC22B interacted with lipid transfer proteins, a factor which, when inhibited, has been shown to result in changes in lipid metabolism and transfer [44].

Simultaneously, four proteins critical to mitochondrial function were also upregulated in the gray matter. The mitochondrial localized enzyme, acetyl-CoA acetyltransferase 1 (ACAT1), has been linked to cholesterol homeostasis and metabolism [72,73] and can be found in axons of the cerebral cortex and hippocampus [49]. Dynamin 1-like (DNM1L) is critical to mitochondrial fission [74]. This protein drives mitochondrial division by self-assembling into filaments that constrict around the organelle [75]. The 2′,3′-cyclic nucleotide 3′-phosphodiesterase (CNP) can also be found in the inner membranes of mitochondria [76] and is important for Ca^2+^ transport [77]. Interestingly CNP levels decreased in non-synaptic mitochondria in the brains of old rats [77]. An increase in CNP release suggests mitochondria may be in a calcium-overloaded condition [78]. Inner membrane mitochondrial protein (IMMT), also known as MIC60 and mitofilin, is important for protein translocation across the mitochondrial membrane, regulating both morphology and protein biogenesis [79].

Even though the tissues were collected and prepared for analysis less than five hours after landing, it is possible that the changes in proteins related to metabolism are simply a response to the landing and do not reflect changes in the spaceflight environment. To confirm or deny this possibility would require that mice be euthanized in orbit and tissues immediately preserved for analysis on the ground. Indeed, such studies have already been planned. However, in our previous study examining the livers of these same mice, there were significant changes in lipid metabolism indicative of a pre-diabetic state that seems to suggest a long-term effect rather than an acute response due to landing [32,59,80].

### 3.4. Oxidative Stress and Tissue Damage Responses

The changes noted in metabolism are likely related to increases in proteins involved with oxidative stress and inflammation in the white matter. Arginase 1 (ARG1), which was highly upregulated, is an important enzyme of the urea cycle that is generally found in the liver and is critical to removing ammonia from the body [81]. However, ARG1+ is also commonly expressed by alternatively-activated macrophages and microglia [82], which tend to be anti-inflammatory. In the brain, ARG1+ microglia have been implicated in amyloid beta plaque removal [83]. Furthermore, it is important to nitric oxide (NO)-mediated vasodilation in microvascular endothelial cells [84]. In activated macrophages, ARG1+ competes with NO synthase (NOS) for their common substrate, l-arginine, leading to a reduction in NO production [85].

Not surprisingly, many of the mitochondrial proteins that were upregulated in the gray matter are also involved in the oxidative stress response. ACAT1 expression has been shown to be elevated during conditions of oxidative stress [49]. Mutations in DNM1L often result in death associated with oxidative stress-induced neurodegeneration [75,86]. In vitro, deleting DNML1 from Purkinje cells resulted in increased oxidative damage via the peroxidation of proteins and lipids [75]. Decreases in IMMT appear to result in increases in ROS levels [87] and seem to be an anti-apoptotic protein important for the regulation of cytochrome c release [87,88,89]. Interestingly, the oxidative damage due to lipid peroxidation found in the interfibrillar mitochondria of diabetic heart tissue was reduced by overexpressing IMMT [90,91].

Increases in the above oxidative stress factors are consistent with the decreases noted in another mitochondrial protein, UQCRB (ubiquinol-cytochrome c reductase binding protein), a subunit of complex III in the mitochondrial respiratory chain [92] of the gray matter. UQCRB is important in mediating mitochondrial-derived reactive oxygen species (ROS) that are both independent of NADPH oxidase and important for angiogenesis [93]. Drugs which inhibit the activity of UQCRB reduce the ROS produced by the mitochondria [92,93,94].

Although not found in mitochondria, Quinonoid dihydropteridine reductase (QDPR) is also upregulated in the gray matter. QDPR is primarily associated with the regeneration of tetrahydrobiopterin (BH4) from quinonoid dihydrobiopterin (qBH2). This is important because BH4 is a critical co-factor in the generation of all three NO synthases, iNOS, nNOS, and eNOS [95]. In quinonoid dihydropteridine reductase (QDPR)^−/−^ knockout mice, the biomarkers of folate-dependent oxidative stress such as ophthalamate, spermine, and γ-Glu-Cys all appeared to be elevated [95].

Given the changes in markers indicative of oxidative stress, it should not be surprising that there were also changes in proteins related to cell damage and death. In the gray matter, at least four upregulated proteins dealing with intracellular damage and/or cell death appeared to be involved. This is consistent with the IPA analysis that found changes in proteins related to organismal death, degeneration of cells, and neurodegeneration. ATP6V0A1 is critically important in mediating autophagosome-lysosome fusion [96]. DNAJC5 appears to have some influence on protein folding and endosomal autophagy, depending on the presence of SGT and Hsc70 [97]. ACAT1 is instrumental in induction of necroptosis through lipid droplet formation that has been demonstrated to be the initial key event in cell death [98]. Finally, CNP appears to play a role with caspase-independent apoptosis [99].

### 3.5. Activation of Catecholamines

Finally, there were several changes in the white matter involved in sympathetic activity and catecholamine production. As mentioned previously, CACNA2D1 was upregulated in the white matter. Although we did not specifically look at areas within the brain, this protein is active in the periventricular nucleus (PVN) and is involved with sympathetic outflow [13]. This is interesting because two proteins involved with sympathetic responses were also upregulated in the gray matter. Quinonoid dihydropteridine reductase (QDPR) is primarily associated with the regeneration of tetrahydrobiopterin (BH4) from quinonoid dihydrobiopterin (qBH2). BH4 is a critical co-factor in the biosynthesis of the neurotransmitters dopamine and serotonin [95]. Syntaxin 1A (STX1A) is expressed in endocrine cells [18] and STX1A knockout mice had decreased circulating levels of the stress hormones, CRH and ACTH, as well as serotonergic precursors [18].

## 4. Materials and Methods

### 4.1. STS-135 Flight Mice and Control Conditions

Space Shuttle Atlantis, i.e., Space Transportation System 135 (STS-135), was launched from the Kennedy Space Center (KSC) on a 13-day mission in July of 2011. Female C57BL/6 mice (Charles River Laboratories, Inc., Hollister, CA, USA) were flown in STS-135 using NASA’s animal enclosure modules (AEMs). Mice were housed in two groups of five per AEM, separated by mesh wire. A set of ground controls (Ground AEMs) was housed at the Space Life Science Laboratory (SLSL) at the KSC. Ground AEM control mice were placed into the same housing hardware used in flight and environmental parameters such as temperature and carbon dioxide (CO_2_) levels were matched as closely as possible based on 48-h delayed telemetry data.

All mice were under ambient temperatures of 26–28 °C with a 12-h day/night cycle during the flight. The mid deck CO_2_ levels that the mice were exposed to averaged 2150 parts per million (ppm) and ranged from a few hundred ppm while on the ground, before installation in the shuttle, to a maximum level of 3480 ppm in the shuttle during the mission. AEM controls were fed a special NASA food bar diet, the same as the space-flown mice. All mice received the same access to food and water *ad libitum*.

The Loma Linda University (LLU) Institutional Animal Care and Use Committee (IACUC) was consulted but no protocol was required since tissues were only obtained after euthanasia. However, it should be noted that all NASA research with vertebrate animals is done in strict accordance with the recommendations in the Guide for the Care and Use of Laboratory Animals of the National Institute of Health (NIH). The primary science team responsible for running the project obtained approval from the NASA Ames Research Center ACUC (NAS-11-002-Y1) on 5/31/2011.

Upon return to Earth, animals were removed from the AEM nursing facility and assessed for survival and health. It was reported that all the mice survived the 13-day space mission. All animals were described by the inspecting personnel as being in good condition.

### 4.2. Dissection and Preservation of Mouse Brains Post Flight

Within 3–5 h after landing, the mice were euthanized and the brains were removed (4–8 mice/group). As part of the primary science, all mice underwent dual energy X-ray absorptiometry (DEXA) densitometry (Piximus, Inc., Fitchburg, WI, USA) immediately prior to anesthesia and euthanasia. Mice were anesthetized with 3–5% isoflurane and euthanized with 100% CO_2_ and exsanguination via cardiac puncture. The whole brain hemispheres were fixed in 4% paraformaldehyde in phosphate buffered saline (PBS) for 24 h and then rinsed with PBS and infiltrated overnight with 30% sucrose in PBS at 4 °C. Fixed samples were sent to Loma Linda University (LLU) via courier for analysis. As we were part of the NASA Biospecimen Sharing Program (BSP) team, we did not have access to all the tissues from all the mice. However, we received tissues from four flight mice and eight ground control mice. Tissue from two of the ground control samples were lost during processing, giving us a final sample size of *n* = 4 or 6 for flight and ground, respectively.

### 4.3. Brain Sectioning for Proteomic Analysis

Paraffin-embedded brain sections were coronally gross sectioned at a thickness of 10 μm. After mounting and de-paraffinizing, six 3 mm diameter circular punches of tissue were obtained to provide three replicate samples each of white matter of the corpus callosum and gray matter of the cerebral cortex. Each tissue punch (disk) was transferred to a vial for individual processing (work up) by a modification of the methodology of Craven et al. [100]. When tissues are processed by the Craven method, formalin fixed tissues still yield 80% or more of the available proteins for proteomic analyses [100]. The preparation of the tissue samples for proteomic mass spectrometry (MS) analysis in this paper is described by the steps below.

### 4.4. Protein Extraction from Tissue

We used boil-proof 1.7 mL low retention polypropylene snap-top vials to process the tissues. Lysis buffer was made by combining 50 μL each of stock solutions A and B, plus 10 μL of freshly prepared 1.1 M dithiothreitol (DTT). Solution A consisted of 300 mM Tris-Cl pH 8.2. Solution B was made by combining five liquid components: 20% sodium dodecyl sulfate (SDS), glycerol, trifluorethanol, thiodiethanol, and water at the volumetric ratio of 3:2:1:1:3. Protein extraction was initiated by the addition of 110 μL lysis buffer to each vial containing a tissue punch (disk), followed by incubation for 30 min at 105 °C in a fume hood because of trifluorethanol toxicity. After cooling the vials were stored indefinitely at −80 °C.

### 4.5. Trypsin Digestion

This was done using Amicon Ultra-0.5 mL 30 kDa centrifugal filter devices (http://www.emdmillipore.com). A urea-ammonium bicarbonate buffer (UA) that is 50 mM with respect to ammonium bicarbonate (ABC) and 8 M with respect to ultrapure urea (https://www.thermofisher.com) was prepared fresh on the day of use. The use of the Amicon Ultra centrifugal filter device has been outlined next. (a) To each vial after protein extraction (see above) 300 μL of UA were added and vortexed. The sample components were transferred to an Amicon Ultra, centrifuged at 10,000× *g* for 10 min, and the filtrate was discarded. (b) To complete the transfer initiated above and to continue the replacement of reagents from the protein extraction procedure described in 2.4 above with UA constituents, the above process was repeated twice using 400 μL UA each time, and both filtrates were discarded. (c) Next, 5 μL each of 5 mM Tris (2-carboxyethyl) phosphine (TCEP) and 10 mM acrylamide and 200 μL UA were combined. The entire volume was then transferred to the Amicon Ultra, incubated for 30 min at ambient temperatures, centrifuged, and the filtrate was discarded. (d) The reagents were washed by the addition of 400 μL UA, centrifuged, and the filtrate was discarded. (e) The urea was washed out by the addition of 400 μL of 50 mM ABC, centrifuged, and the filtrate was discarded. This process was repeated twice more. (f) Next, 200 μL of 50 mM ABC were added and a protein assay (Nanodrop, http://www.nanodrop.com, or Micro BCA, http://www.thermofisher.com) was performed on an aliquot. (g) A prepared trypsin digestion solution was added to the remaining protein in the Amicon Ultra. The 100 μL of 250 mM ABC were combined with 2.5 μL of 1% Promega PMax (http://promega.com) to give a 0.025% working solution plus sufficient sequencer-grade trypsin to result in an enzyme to protein ratio in the range of 1:25–50. (h) The solution was incubated overnight at 37 °C to convert the proteins to peptides. (i) The next morning, 150 μL of water was added, centrifuged, and the filtrate containing the peptides was collected. These steps were repeated using another 150 μL water and the filtrates were combined. (j) Next, 20 µL of 10% TFA were added to the combined filtrates to destroy the PMax. (k) The filtrate was concentrated in a vacuum centrifuge to a volume of 100 μL and stored at −80 °C. (l) Just prior to MS analysis, a 2–5 μg portion of each protein/peptide sample was purified with a Zip-tip C18 P10 (http://www.emdmillipore.com). The sample preparations from each set of the three replicate “punches” were pooled, resulting in a total of 20 samples for MS analysis.

### 4.6. MS Analyses and Data Processing

An Easy-nLC system with an autosampler was attached to an LTQ-Orbitrap Velos Pro mass spectrometer (http://thermoscientificbio.com). This was used for all MS analyses utilizing 2 to 2.5 µg peptide loadings after zip-tip purification. After injection, the 5 µL peptide samples in 0.25% TFA were passed through a 2 cm × 100 μm C18, of 5 μm particle size, precolumn (http://thermoscientificbio.com) in series with a 10 cm × 75 μm in-house-prepared capillary column packed with Microm Magic C18 of 5 μ particle size (http://www.michrom.com/) for separation and elution.

A 2-h gradient was used (Solvent A 0.1% FA in water, Solvent B 0.1% FA in ACN, 5–30% Solvent B). Collision-induced disassociation was used to fragment the top 10 most abundant ions and the MS/MS spectra were collected between 250 and 1500 *m*/*z*, following the parent full-scan mass spectrum collected at 60,000 resolution.

The raw data files from these analyses were processed first through Proteome Discoverer (http://thermoscientificbio.com) for precursor intensity with the following search parameters using the mouse Uniprot/Swissprot database: 2 missed trypsin cleavages allowed, dynamic oxidation on methionine, deamidation on asparagine and glutamine, and static proprionamide attachment on cysteine. In parallel, the data processing was performed using our DeMix Q algorithm. Briefly, DeMix “unmixes” the MS/MS events that become frequently but accidentally multiplexed because more than one (up to four) precursor ions are selected for fragmentation within the same *m*/*z* window simultaneously [101]. DeMix allows one to identify more peptides on average than there were precursors selected for MS/MS. For extension of the peptide identity “between the runs” and thus additionally increase the number of identified peptides while simultaneously improving quantification accuracy, we used DeMix Q [102], an extension of the DeMix algorithm that employs the scoring of identity transfer. Further improvement in quantification accuracy can be achieved with the help of our Diffacto algorithm [103] that selects for protein quantification of only “well-behaved” peptides.

All spread sheets with protein relative abundances were then exported into Ingenuity Pathway Analysis (IPA; http://www.ingenuity.com/products/ipa). In IPA, only direct relationships were allowed in the general settings and genes. Endogenous chemical interactions and causal networks were both included. All items were selected under data sources, confidence, species and mutation.

## 5. Conclusions

In summary, this study is the first to identify spaceflight-induced proteomic significance and biomarkers in the gray and white matter of the murine brain. These unique “proteomic signatures” of brain tissue may provide new mechanistic insight into the complex biological response to space environment. We propose that spaceflight conditions induce changes in neuronal structure, cellular function, oxidative response, mitochondrial function and metabolism, which, in turn, might lead to tissue injury and late neurodegeneration. The diverse changes in protein expression profiles in white and gray matter in response to spaceflight conditions warrants further investigation. Further studies are also necessary to elucidate the possible tissue and functional impact responsible for our findings and to identify effective countermeasures.

## Figures and Tables

**Table 1 ijms-20-00007-t001:** Spaceflight induced alterations to the proteomic profile in the white matter.

Symbol	Entrez Gene Name	Expr *p*-Value	Expr Log Ratio	Location	Type(s)
ARG1	arginase 1	0.037	3.742	Cytoplasm	enzyme
CACNA2D1	calcium voltage-gated channel auxiliary subunit alpha2delta 1	0.024	2.244	Plasma Membrane	ion channel
PPFIA3	PTPRF interacting protein alpha 3	0.025	2.189	Plasma Membrane	phosphatase
PITPNA	phosphatidylinositol transfer protein alpha	0.010	1.328	Cytoplasm	transporter
MYO5A	myosin VA	0.034	1.304	Cytoplasm	enzyme
DYNLL2	dynein light chain LC8-type 2	0.040	1.152	Cytoplasm	other
VPS35	VPS35, retromer complex component	0.046	−1.164	Cytoplasm	transporter
GAPDH	glyceraldehyde-3-phosphate dehydrogenase	0.015	−1.478	Cytoplasm	enzyme
MBP	myelin basic protein	0.003	−2.536	Extracellular Space	other

**Table 2 ijms-20-00007-t002:** Spaceflight induced alterations to the proteomic profile in the gray matter.

Symbol	Entrez Gene Name	Expr *p*-Value	Expr Log Ratio	Location	Type(s)
QDPR	quinoid dihydropteridine reductase	0.0229	2.458	Cytoplasm	enzyme
DNM3	dynamin 3	0.00186	1.781	Cytoplasm	enzyme
ACAT1	acetyl-CoA acetyltransferase 1	0.0000214	1.637	Cytoplasm	enzyme
DNAJC5	DnaJ heat shock protein family (Hsp40) member C5	0.00702	1.501	Plasma Membrane	other
SH3GL2	SH3 domain containing GRB2-like 2, endophilin A1	0.0136	1.377	Plasma Membrane	enzyme
RAP1GDS1	Rap1 GTPase-GDP dissociation stimulator 1	0.0284	1.363	Cytoplasm	other
DNM1L	dynamin 1-like	0.0069	1.284	Cytoplasm	enzyme
CNP	2′,3′-cyclic nucleotide 3′ phosphodiesterase	0.000816	1.261	Cytoplasm	enzyme
YWHAE	tyrosine 3-monooxygenase/tryptophan 5-monooxygenase activation protein epsilon	0.00989	1.261	Cytoplasm	other
ACTN1	actinin alpha 1	0.0308	1.254	Cytoplasm	transcription regulator
ATP6V0A1	ATPase H+ transporting V0 subunit a1	0.00028	1.191	Cytoplasm	transporter
IMMT	inner membrane mitochondrial protein	0.00436	1.154	Cytoplasm	other
NEFL	neurofilament light	0.00557	1.153	Cytoplasm	other
ENO2	enolase 2	0.0281	1.123	Cytoplasm	enzyme
STX1A	syntaxin 1A	0.00859	1.074	Cytoplasm	transporter
UQCRB	ubiquinol-cytochrome c reductase binding protein	0.0447	−1.059	Cytoplasm	enzyme
SEC22B	SEC22 homolog B, vesicle trafficking protein (gene/pseudogene)	0.00601	−2.709	Cytoplasm	other

**Table 3 ijms-20-00007-t003:** Spaceflight induced changes in white and gray matter proteins indicative of functional deficits.

Region	Categories	Diseases or Functions Annotation	*p*-Value	Activation *Z*-Score
**White Matter**	Cell Morphology, Cellular Assembly and Organization, Cellular Function and Maintenance	formation of cellular protrusions	1.69 × 10^−3^	−1.982
	Organismal Survival	organismal death	3.76 × 10^−2^	−1.156
	Cellular Assembly and Organization, Cellular Function and Maintenance	organization of cytoplasm	6.40 × 10^−5^	−1.154
	Cellular Assembly and Organization, Cellular Function and Maintenance	microtubule dynamics	9.47 × 10^−5^	−1.154
	Tissue Morphology	quantity of cells	3.44 × 10^−2^	−0.44
	Cell Death and Survival	apoptosis	4.26 × 10^−2^	0.003
	Cell Death and Survival	necrosis	1.41 × 10^−2^	0.014
	Cell Death and Survival	cell death of tumor cell lines	4.40 × 10^−3^	0.028
	Cell Death and Survival	cell death	1.41 × 10^−2^	0.166
	Cell Death and Survival	apoptosis of tumor cell lines	3.86 × 10^−2^	0.176
	Lipid Metabolism, Molecular Transport, Small Molecule Biochemistry	concentration of lipid	7.47 × 10^−3^	0.333
**Gray Matter**	Organismal Survival	organismal death	1.22 × 10^−3^	−3.257
	Organismal Injury and Abnormalities	organ degeneration	1.46 × 10^−5^	−2.186
	Cellular Compromise	degeneration of cells	1.36 × 10^−4^	−1.982
	Developmental Disorder	growth failure	1.45 × 10^−2^	−1.982
	Neurological Disease	neurodegeneration	8.36 × 10^−5^	−1.981
	Cell Death and Survival	necrosis	1.84 × 10^−5^	−1.6
	Cell Death and Survival	apoptosis	5.78 × 10^−3^	−1.404
	Cell Death and Survival	neuronal cell death	2.96 × 10^−3^	−0.958
	Molecular Transport	transport of molecule	9.80 × 10^−3^	−0.722
	Cell Death and Survival	cell viability	1.48 × 10^−2^	0.555
	Infectious Diseases	infection by HIV-1	2.45 × 10^−3^	1.98
	Infectious Diseases	Viral Infection	1.12 × 10^−2^	2.2

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
