# Peer review of "Proteomic Analysis of Mouse Brain Subjected to Spaceflight"

_ijms, 2018, doi:10.3390/ijms20010007_

Round 1

Reviewer 1 Report

The paper by Xiao Wen Mao and co-authors entitled “Proteomic analysis of mouse brain subjected to spaceflight” describes the effects of 13-days long spaceflight on the proteins expression.

The major points of criticism are following.

There is a series of studies describing the effect of long-term 30-days long spaceflight on the expression of several neurogenes in C57Bl/6 mice. PMID numbers are 25944479, 25451288, 25084757. Authors should mention these papers in the introduction as well as to take these data into account and discuss it in comparison with the own data.

Also authors should add more details about mice subjected to the spaceflight and control mice in the Materials and Methods section. Does cages for spaceflight and control animals were the same? What was a time span between spacecraft landing and animals sacrifice? Is it was enough for fast-response proteins to restore to a base level?

Authors should explain their choice of animal sex. Usually, mouse males are used in experiments to avoid the effect of the estrous cycle.

I believe that the MS will benefit if authors will add information (if it is possible) on structure-specific protein expression. Otherwise, authors should explain why they analyzed whole-brain slices but not brain structures.

There are a lot of data on the effects of simulated microgravity on the different genes and proteins. I think that authors should use these data for comparison with their own results and add this comparison in the discussion section.

Author Response

Reviewer 1

Comment 1: There is a series of studies describing the effect of long-term 30-days long spaceflight on the expression of several neurogenes in C57Bl/6 mice. PMID numbers are 25944479, 25451288, 25084757. Authors should mention these papers in the introduction as well as to take these data into account and discuss it in comparison with the own data.

Response: While it is certainly true that there are several papers describing changes in gene expression after both spaceflight and ground based models (including the three listed), we purposely tried to limit the discussion to proteomics. This is primarily because we have another gene expression dataset from the cerebellum of these same mice and that will likely be put into a separate manuscript.  Furthermore, the manuscript is already rather lengthy (20 pages with 100 references), and we felt that including additional discussion involve gene expression in the present paper would make the whole thing unwieldy and difficult to read. (Note: We apologize for how long some of this work is taking to publish, but a quick pubmed search of many of the authors of this manuscript will indicate that there has been a steady flow of publications from these animals over the last few years.)

Comment 2: Also authors should add more details about mice subjected to the spaceflight and control mice in the Materials and Methods section. Does cages for spaceflight and control animals were the same? What was a time span between spacecraft landing and animals sacrifice? Is it was enough for fast-response proteins to restore to a base level?

Response: The methods section already states that “Female C57BL/6 mice (Charles River Laboratories, Inc., Hollister, CA) were flown in STS-135 using NASA’s animal enclosure modules (AEMs). A set of ground controls (Ground AEMs) was housed at the Space Life Science Laboratory (SLSL) at KSC. Ground AEM control mice were placed into the same hardware used in flight and environmental parameters such as temperature and carbon dioxide (CO2) levels were matched as closely as possible based on 48-hour delayed telemetry data.” However, to make this statement even clearer, we added the term “housing” to the highlighted phrase above so that it now reads, “Ground AEM control mice were placed into the same housing hardware used in flight…”

As for the timing between landing and sacrifice, the methods section also already states, “Within 3-5 hours after landing, the mice were euthanized and brains were removed (4-8 mice/group).”  It is quite possible that some proteins have returned to baseline levels in this time frame, particularly for those proteins that may be involved in inflammatory responses (such as cytokines), which are tightly regulated in the brain. The only way to resolve this issue is to perform the same test on brains isolated from mice euthanized and dissected in orbit.  We may have the chance to do so on a future ISS study (date TBD, likely in 2019), but the conditions for that experiment will not be identical to the space shuttle experiment described in this paper (e.g. different hard ware, different length of time in orbit, no fixation, etc.).

Comment 3: Authors should explain their choice of animal sex. Usually, mouse males are used in experiments to avoid the effect of the estrous cycle.

Response: The choice of the gender of the mice was out of our control.  We collected tissues as part of a NASA’s Biospecimen Sharing Program (BSP).  The primary investigation was under the control of investigators at Amgen and BioServe Space Technologies.  Also, at the time of the flight (2011), all flights that involved rodents required that female mice be used to avoid the risk of stress-induced aggression and fighting, which is common with C57BL6 mice. Finally, most of our preliminary ground-based work involved female mice.

Comment 4: I believe that the MS will benefit if authors will add information (if it is possible) on structure-specific protein expression. Otherwise, authors should explain why they analyzed whole-brain slices but not brain structures.

Response: We did not analyze whole brain. But rather, we collected and pooled three samples from the cortical region (gray matter) or three from the corpus callosum (white matter) using 3 mm diameter circular punches in a mouse brain. Due to the small size of the brain and necessary amount of tissue required for our protocol, it is not feasible to select sub region/structures within these regions, especially for corpus callosum which is a very narrow band under the cortex.  We have added the location information to the manuscript to clarify this process.

Comment 5: There are a lot of data on the effects of simulated microgravity on the different genes and proteins. I think that authors should use these data for comparison with their own results and add this comparison in the discussion section.

Response: See response to Comment 1 above.

Reviewer 2 Report

In this manuscript  the Authors aimed to investigate the possible mechanism/s induced by space environment on brain protein expression profiles from  mice that have experienced a 13-day spaceflight. The research field regarding the effect of microgravity, or more, space environment on living beings implies not only expertise in the experimental procedures, but also some challenges to be overcame and some limits, as: - high costs, - very often the carrying out of the experiments and data collection are possible only at the end and not during the exposition, -  the remote possibility to replicate the study at the same experimental conditions above all in real, more than in simulated, space environment. The corresponding Author and most co-authors have a multi-year experience in this field and the study is well documented.

Anyway in my opinion, some points have to be better elucidated and discussed.

1- The Results (in which a subtitle is pleonastic because there is only one paragraph) are shortly described above all regarding table 3 in which cellular functions appears generic and not brain-tissue specific as synaptic plasticity, mentioned at the beginning of the Discussion. This last one appears too much detailed and a list of down- and up-regulated proteins in white and grey matter, in which the reader is lost. In addition, as the main findings regard two aspects: the neuronal structure/function and the metabolism, the Discussion could be divided into two paragraphs concerning these aspects.  

2- In the Results, the Authors stated that “In the white matter, there was a strong trend for a down-regulation in functions related to the formation of cellular protrusions (Z = -1.98)” (lines 167-168); while at the beginning of the Discussion they reported that “In the white matter, there were three proteins related to synaptic plasticity that were upregulated. This is consistent with the upregulation in proteins related to the “formation of cellular protrusions” found in the IPA analysis (Table 3) (lines 181-183)…. It appears confusing and contradictory. Please explain better.

3- In the study, the mice were female but I did not find their age, can the Authors explain the reason of their choice? There are some published evidence regarding differences between male and female brain in mice (for example see: Guneykaya et al, Cell Rep 24:2773-2783, 2018; Block et al, Biology of Sex Differences 6:24, 2015), the Authors could also mention or discuss this point in the Discussion.

4- Tissue changes related to structures are consistent with effect of a long exposition time (as 13 days), but changes in metabolic pathways could be related also to acute event (or at least this can not be excluded) as that at the moment of return to Earth and landing. The Authors could also mention or discuss this point in the Discussion.

Author Response

Reviewer 2

Comment 1: In this manuscript the Authors aimed to investigate the possible mechanism/s induced by space environment on brain protein expression profiles from mice that have experienced a 13-day spaceflight. The research field regarding the effect of microgravity, or more, space environment on living beings implies not only expertise in the experimental procedures, but also some challenges to be overcame and some limits, as: - high costs, - very often the carrying out of the experiments and data collection are possible only at the end and not during the exposition, -  the remote possibility to replicate the study at the same experimental conditions above all in real, more than in simulated, space environment. The corresponding Author and most co-authors have a multi-year experience in this field and the study is well documented.

Response: We thank Reviewer 2 for the positive comments, e.g. that studies such as this do require expertise in experimental procedures, experienced authors, and good study documentation.

Comment 2: The Results (in which a subtitle is pleonastic because there is only one paragraph) are shortly described above all regarding table 3 in which cellular functions appears generic and not brain-tissue specific as synaptic plasticity, mentioned at the beginning of the Discussion.

Response: We have removed the subtitle from the results section.  We apologize, but this was an artifact left over from when we had originally thought to include our gene expression data from these same mice (see the response to Comment 1 from Reviewer 1).

We agree that the cellular functions listed in Table 3 are generic cellular functions. We included them, as described in the manuscript, because these functions have been identified by the IPA software as being significantly impacted by the spaceflight environment.

Presumably, by using the term “brain-tissue specific,” the reviewer is referring to neurons.  However, as you know, not all cells in the brain are neurons and other cell types such as microglia, astrocytes, and all the cell types involved in the brain vasculature, also play significant roles in brain function. Furthermore, these non-neuronal cell types in the brain may actually be more responsive to the spaceflight environment than the neurons themselves (e.g. immune populations like microglia are more susceptible to increases in radiation than the typical neuron).

Even if this were not the case, the spaceflight environment includes slightly increased levels of radiation (for a discussion, see Radiat Prot Dosimetry. 2005;116(1-4 Pt 2):374-9) and slightly elevated levels of carbon dioxide within space craft (for an example on the ISS, see J Occup Environ Med. 2014 May;56(5):477-83), not to mention psychological stress and other factors. Because of this fact, we believe that normal cellular function in ALL cell types should be considered when evaluating the response to spaceflight. In our opinion, focusing only on neurons would be insufficient. 

Comment 3: This last one appears too much detailed and a list of down- and up-regulated proteins in white and grey matter, in which the reader is lost. In addition, as the main findings regard two aspects: the neuronal structure/function and the metabolism, the Discussion could be divided into two paragraphs concerning these aspects. 

Response: By “This last one…,” we believe that the reviewer is referring to the Discussion in general. We fully agree that it is a difficult discussion.  We struggled with how to write it for quite a while because there were no clear and obvious changes in protein pathways as one would see in a disease state or after brain injury.

To help with writing the discussion, we originally had subtitles spread throughout the discussion but ended up deleting them in the final edit. The reviewer rightly identified two major themes in the data.  However, by our count, there were five. However, perhaps this editorial decision was premature and upon reflection, we have added the subtitles back to the discussion. Hopefully they will guide the reader as they guided us when we were organizing and writing the manuscript.

Comment 4: In the Results, the Authors stated that “In the white matter, there was a strong trend for a down-regulation in functions related to the formation of cellular protrusions (Z = -1.98)” (lines 167-168); while at the beginning of the Discussion they reported that “In the white matter, there were three proteins related to synaptic plasticity that were upregulated. This is consistent with the upregulation in proteins related to the “formation of cellular protrusions” found in the IPA analysis (Table 3) (lines 181-183)…. It appears confusing and contradictory. Please explain better.

Response: We apologize and thank the reviewer for pointing out this obvious error. We have deleted the first quoted sentence from the discussion. Upon closer examination, the relevant proteins noted to be involved in this function by IPA are: MBP, MYO5A, OPA1, PALM and PITPNA. Though downregulated, two of these proteins (OPA1 and PALM) did not meet our criteria for significance (i.e. both had log fold changes less than 1). The other three have additional functions that are unrelated to dendrite formation. It is likely that this functional change was more related to other cells (i.e. glial cells) and is not relevant to the discussion of dendrites.

Comment 5: In the study, the mice were female but I did not find their age, can the Authors explain the reason of their choice? There are some published evidence regarding differences between male and female brain in mice (for example see: Guneykaya et al, Cell Rep 24:2773-2783, 2018; Block et al, Biology of Sex Differences 6:24, 2015), the Authors could also mention or discuss this point in the Discussion.

Response: As indicated in our response to Reviewer 1, the choice of the gender of the mice was out of our control.  We collected tissues as part of a NASA’s Biospecimen Sharing Program (BSP).  The primary investigation was under the control of investigators at Amgen and BioServe Space Technologies.  Also, at the time of the flight (2011), all flights that involved rodents required that female mice be used to avoid the risk of stress-induced aggression and fighting, which is common with C57BL6 mice. Finally, most of our preliminary ground-based work involved female mice.

Comment 6: Tissue changes related to structures are consistent with effect of a long exposition time (as 13 days), but changes in metabolic pathways could be related also to acute event (or at least this can not be excluded) as that at the moment of return to Earth and landing. The Authors could also mention or discuss this point in the Discussion.

Response: The reviewer makes an excellent point and it is one we’ve discussed in previous manuscripts from these mice involving other tissues.  Therefore, we have added the following paragraph to the discussion to address this issue:

Even though the tissues were collected and prepared for analysis in less than 5 hours from landing, it is possible that the changes in proteins related to metabolism are simply a response to the landing and do not reflect changes in the spaceflight environment. To confirm or deny this possibility would require that mice be euthanized in orbit and tissues immediately preserved for analysis on the ground.  Indeed, such studies have already been planned. However, in our previous study examining the liver from these same mice, there were significant changes in lipid metabolism that seem to suggest a long-term effect rather than an acute response due to landing [refs included in the manuscript].

Reviewer 3 Report

The present paper reports the effects of spaceflight on the mouse brain. The proteomic analysis showed that 13-day spaceflight altered the expression of 26 proteins in the gray matter and white matter. Although the same proteins were not changed in the gray and white matters, the affected proteins in both regions were functionally closely related. The present study provided interesting and valuable results including for the astronauts’ health, but there are some methodological points to be addressed.

1.   Why were female mice used? Female mice have sex cycle which affects the structure and function of neurons, including structure of dendritic spines. Was sex cycle checked in mice of the spaceflight and the ground control?

2.   It was explained that 4-8 mice/group were used. Please explain the exact number of the experimental and control mice.

3.   The condition in the spaceflight is not clear. Were mice kept in a group in the AEM, or individually housed?

4.   Brains were fixed in 4% paraformaldehyde. Are there any effects in the proteomic analysis, using the fixed sample?

5.   Brains were frozen in OCT compound (lines 90-91). But they were deparaffinized (lines 93-94). Which is correct?

6.   The critical point is that the brain regions in the white and gray matters are not explained clearly. The structure and function of the brain are different, depending on the brain regions.

Minor points

What are ??? in line 101?

Both “grey” (e.g. lines 18, 19, 42) and “gray”(e.g. lines 58, 95)  matters are used. 

Author Response

Reviewer 3

Comment 1: The present paper reports the effects of spaceflight on the mouse brain. The proteomic analysis showed that 13-day spaceflight altered the expression of 26 proteins in the gray matter and white matter. Although the same proteins were not changed in the gray and white matters, the affected proteins in both regions were functionally closely related. The present study provided interesting and valuable results including for the astronauts’ health, but there are some methodological points to be addressed.

Response: We thank the reviewer for the positive response to the work.

Comment 2: Why were female mice used? Female mice have sex cycle which affects the structure and function of neurons, including structure of dendritic spines. Was sex cycle checked in mice of the spaceflight and the ground control?

Response: We did not check the estrus cycle of the mice. It is possible that other Biospecimen Sharing Program investigators (e.g. Dr. Joseph Tash) did so as part of their work with the same mice.  However, we do not have access to their data and, as far as we know, it has not yet been published (at least not on a mouse by mouse basis). 

In any case, as indicated in our response to Reviewer 1, the choice of the gender of the mice was out of our control.  We collected tissues as part of a NASA’s Biospecimen Sharing Program (BSP).  The primary investigation was under the control of investigators at Amgen and BioServe Space Technologies.  Also, at the time of the flight (2011), all flights that involved rodents required that female mice be used to avoid the risk of stress-induced aggression and fighting, which is common with C57BL6 mice. Finally, most of our preliminary ground-based work involved female mice.

Comment 3: It was explained that 4-8 mice/group were used. Please explain the exact number of the experimental and control mice.

Response: As mentioned above, we were part of the BSP team for this flight so we didn’t have control of the tissues from all the mice. Therefore, we only got a subset of the available tissues. The original number of mice intended for this project was, indeed 4 flight and 8 ground.  However, two of the ground control samples were lost during processing.  So we ended up with 4 flight and 6 ground samples that were actually analyzed. We did this for both white and gray matter. We have amended the methods section to reflect this.

Comment 4: The condition in the spaceflight is not clear. Were mice kept in a group in the AEM, or individually housed?

Response: The AEM is a group housed facility. We have added the following sentence to the methods section to clarify this: 

Mice were house in 2 groups of 5 per AEM, separated by a wire mesh.

Comment 1: 4.   Brains were fixed in 4% paraformaldehyde. Are there any effects in the proteomic analysis, using the fixed sample?

Response: When tissues are processed by the Craven method, as cited in the manuscript, formalin fixed tissues still yield 80% or more of the available proteins for proteomic analyses. We have added a statement to the methods indicating this.

Comment 1: 5.   Brains were frozen in OCT compound (lines 90-91). But they were deparaffinized (lines 93-94). Which is correct?

Response: We thank reviewer for pointing out this discrepancy. The methods have have been modified as follows:  

Brains were fixed in 4% paraformaldehyde in phosphate buffered saline (PBS) and embedded in paraffin.

Comment 1: 6.   The critical point is that the brain regions in the white and gray matters are not explained clearly. The structure and function of the brain are different, depending on the brain regions.

Response: The specific regions of the white matter and gray matter have been added to the methods as follows:  

After mounting and de-paraffinizing, six 3 mm diameter circular punches of tissue were obtained to provide 3 replicate samples each of white matter of the corpus callosum and gray matter of the cerebral cortex.

Comment 1: What are ??? in line 101?

Response: We apologize and have removed the “???”. This was an artifact of editing. One of the co-authors asked for DTT to be written out and I must have missed removing the question marks.

Comment 1: Both “grey” (e.g. lines 18, 19, 42) and “gray”(e.g. lines 58, 95)  matters are used.

Response: We thank the reviewer for pointing out this discrepancy. Although the word appears to be spelled both ways in the literature, we have changed all spellings to “gray,” the spelling used in Kandel’s “Principles of Neural Science,” Fifth edition.

Round 2

Reviewer 3 Report

The manuscript was revised satisfactorily.